# Zirconia Dental Implants: A Closer Look at Surface Condition and Intrinsic Composition by SEM-EDX

**DOI:** 10.3390/bioengineering10091102

**Published:** 2023-09-20

**Authors:** Alex Tchinda, Augustin Lerebours, Richard Kouitat-Njiwa, Pierre Bravetti

**Affiliations:** Institut Jean Lamour, Université de Lorraine, Faculty of Science, Department of Micro and Nanomechanics for Life, Unités Mixtes de Recherche 7198, 54011 Nancy, Francerichard.kouitat@univ-lorraine.fr (R.K.-N.); pierre.bravetti@univ-lorraine.fr (P.B.)

**Keywords:** zirconia dental implants, composition, surface condition, alumina, toxicity

## Abstract

Modern dental implantology is based on a set of more or less related first-order parameters, such as the implant surface and the intrinsic composition of the material. For decades, implant manufacturers have focused on the research and development of the ideal material combined with an optimal surface finish to ensure the success and durability of their product. However, brands do not always communicate transparently about the nature of the products they market. Thus, this study aims to compare the surface finishes and intrinsic composition of three zirconia implants from three major brands. To do so, cross-sections of the apical part of the implants to be analyzed were made with a micro-cutting machine. Samples of each implant of a 4 to 6 mm thickness were obtained. Each was analyzed by a tactile profilometer and scanning electron microscope (SEM). Compositional measurements were performed by X-ray energy-dispersive spectroscopy (EDS). The findings revealed a significant use of aluminum as a chemical substitute by manufacturers. In addition, some manufacturers do not mention the presence of this element in their implants. However, by addressing these issues and striving to improve transparency and safety standards, manufacturers have the opportunity to provide even more reliable products to patients.

## 1. Introduction

The success of modern dental implantology is based on a set of more or less related first-order parameters, such as the implant surface condition and the intrinsic composition of the material. Indeed, surface conditions have been extensively studied, showing their direct involvement in the osseointegration process. Thus, it is recognized that the surface topography and the chemistry, including the surface energy of a material, condition the biological behavior of the tissues in contact at the bone/implant interface. In addition, the intrinsic composition of the material conditions, to a certain extent, the mechanical and chemical stresses of the material in contact with the physiological environment of a living organism [1]. For this reason, implant manufacturers have been focusing for decades on the search and development of the ideal material combined with an optimal surface finish to ensure the success and durability of their product. Although a wide range of products made of various biocompatible materials is offered, titanium is currently the most widely used due to its biological and mechanical properties [2]. However, the addition of 6% aluminum to improve the mechanical strength and 4% vanadium to increase the ductility of the titanium is a choice that seems to be favored by the brands because of the mechanical reliability of the grade 5 titanium alloy that has been obtained [3]. The addition of aluminum and vanadium appears to be mechanically beneficial and many studies report an increase in Young’s modulus and the high corrosion resistance of this alloy [4]. Nevertheless, other studies report titanium’s corrosive potential, which could imply releases of ions likely to be harmful to health, in this case aluminum and vanadium [5]. This observation has led manufacturers to consider the potential of using zirconia as an alternative material to titanium. Due to its proven biocompatibility and the increasing aesthetic demands of patients, zirconia is increasingly present in the implantology market. In dentistry, zirconia is generally considered a ceramic, but from a physicochemical point of view, it is a metal oxide with ceramic properties characterized by polymorphism and allotropy. Doped forms of zirconia are very often developed by designers to meet specific demands. The most commonly used doping agent in dental surgery is yttrium oxide at a concentration of 2 to 3 mol%, forming 2Y-TZP or 3Y-TZP (yttrium oxide stabilized tetragonal zirconia polycrystals). In reality, the rate of retained tetragonal or quadratic form at room temperature would depend on the amount of yttrium oxide and the size of the zirconia grains. The result is a “partially stabilized” zirconia with much more interesting mechanical properties due to the phase transformation phenomenon [6]. This obtained zirconia consists of a matrix stabilized to the crystals in quadratic form, within which there is a dispersion of small precipitates of zirconia poor in stabilizers remaining in the metastable state in the quadratic form [6]. It is also reported that zirconia absorbs energy, allowing for both increased toughness and increased strength [6,7]. The literature reports excellent biocompatibility of zirconia and an absence of cytotoxicity of zirconia [8]. Many studies suggest that zirconia implant surfaces are unfavorable to bacterial colonization, thus, preventing the formation of bacterial biofilms and consequently peri-implantitis [9]. In contrast to titanium which is subject to a prevalence of peri-implantitis around the implant [10]. A study conducted by my team suggests that anaerobic bacteria of the genus *Desulfovibrio fairfieldensis* can form biofilm colonies on titanium implant discs [11]. Many preclinical studies report faster epithelial and connective tissue maturation processes around zirconia implants compared to titanium [12]. To further investigate the possible influence of zirconia on the human genome, a recent transcriptomic study conducted by our team suggests that zirconia and yttria zirconia have no deleterious influence on gene expression in human bone cells [13]. This observation echoes the study by Covacci et al. [14] which suggests the same observation regarding the absence of genotoxicity of this material.

Chemical inertness is one of the greatest advantages of zirconia implants over titanium implants in that the strong interatomic bonds within this material result in almost no ionic leaching. This aspect explains the absence or low inflammatory infiltrates of cytokine or interleukin observed following contact with zirconia particles [15]. Thus, the use of this material in titanium-allergic patients is a more than judicious alternative. However, zirconia does not add polymetallism, which is often responsible for oral electroplating that can corrode metals in patients with metal restorations. Thus, the evolution towards a “metal-free” patient context increasingly favors the use of zirconia in implantology [16].

Although the advantages of zirconia are both multiple and promising according to the literature, and although commercial regulations are obliging the manufacturers to respect the imposed standards, the brands do not always communicate transparently on the nature of the marketed products. In fact, the lack of consistent and transparent communication from dental implant manufacturers regarding their products is a cause for concern. Both patients and healthcare professionals should have access to accurate information to make informed decisions about the materials used in dental implant procedures. This transparency is crucial to ensure the safety, effectiveness, and long-term success of dental implants. In this age of advanced technology and scientific research, it is imperative that manufacturers provide detailed information about the composition and characteristics of their dental implant materials. This includes specifying the type of zirconia used, the concentration of stabilizers, such as yttrium oxide, and any other relevant details about the manufacturing process. Patients, in particular, need to be aware of the materials used in their dental implants, especially if they have allergies or specific sensitivities. Patients allergic to titanium, in particular, can greatly benefit from the availability of zirconia implants. Knowing that zirconia offers a viable and safe alternative can provide peace of mind and open up more treatment options for those who may have been previously limited by their allergic reactions to titanium. Furthermore, healthcare professionals must have access to comprehensive information about implant materials to ensure they can choose the most suitable options for their patients. The ability to choose between titanium and zirconia implants based on individual needs and a patient’s medical history is essential for providing high-quality dental care. In the quest for transparency, regulatory bodies should continue to refine and enforce guidelines for dental implant manufacturers. These guidelines should cover not only the materials used but also the need for clear and precise labeling, patient information, and post-market surveillance to monitor the long-term performance of implant materials. Ensuring that manufacturers provide detailed information about dental implant materials, and that both patients and healthcare professionals have access to this information, is essential for the continued advancement of dental implant technology and the overall well-being of patients. Transparency in the industry will lead to safer and more effective dental implant procedures, benefiting individuals seeking dental care around the world. Thus, this study aims to compare the surface finishes and intrinsic composition of three zirconia implants from three leading brands. The brands and models of implants studied are listed in the table below (Table 1).

## 2. Materials and Method

### 2.1. Cutting off the Implants

The cuts were made with a micro-saw (Struers^®^ secotom-10, Struers, Copenhagen, Denmark). The samples were obtained by cross-sections of a 4 to 6 mm thickness from the apical area of each implant, and each sample was analyzed by a profilometer and SEM. This allowed the integrity of the implant to be maintained to continue the analyses for other parallel studies with the same implants.

### 2.2. Surface Characterization with Tactile Profilometer

The measurement of the roughness was performed at three different locations on the implant, and then the data were averaged to obtain values representative of the total implant surface. Data was carried out using a contact or tactile profilometer (Bruker, Billerica, MA, USA, DektakXT stylus^®^). The accuracy of this device is of the order of plus or minus 1 nm.

The operation was performed according to ISO 4287:1997 standards with the following settings:Measurement length: 200 μm.Long cut OFF: 0.8 μm/short cut OFF: 0.08 µm.

The measurements were performed 3 times on each sample in a different area of each sample and then averaged.

### 2.3. SEM-EDS Composition Analysis

The zirconia samples underwent a meticulous metallization process in preparation for in-depth analysis. Firstly, a precisely measured 15-nanometer thick layer of carbon was deposited onto their surfaces. This carbon deposition was carried out using specialized equipment, (Safematic Compact Coating unit-010, Safematic, Zizers, Switzerland) operating under controlled pressure for a period of 10 min. Once metallization was completed, the zirconia disks were transferred to a Quanta™ FEG 650 SEM (Quanta, Houston, TX, USA), (Institut Jean Lamour, Nancy, France). This scanning electron microscope was configured to operate at a voltage of 10.00 kV, allowing for high-resolution observation of the metallized samples. This detailed analysis was crucial for examining the nanoscale structure and characteristics of the zirconia samples.

## 3. Results

### 3.1. Surface Roughness

The average surface roughness of the three measurements on zirconia implants Table 2 and Figure 1, Figure 2 and Figure 3 is, contrary to expectations, lower than the average roughness of a Ra between 1 and 3 μm advertised by the manufacturers.

### 3.2. Scanning Electron Microscope Observation

Nobel^®^ reports a specific surface treatment called ZERAFIL™ 6,7 which imparts a hydrophilic property for Nobelpearl. After observation with SEM, in the high magnification ×4000 image (Figure 4), it appears that the surface is covered with micro grains.

On the other hand, the Metoxit^®^ brand Ziraldent implant has a patented zircopore surface treatment. This treatment corresponds to a thermohydrolysis of zirconium salt in solution which allows for obtaining a microporous surface. Indeed, Figure 5 confirms the presence of zirconium grains which at first sight seemed to be the result of sandblasting.

The Z look 3 implant from Z-systems^®^ is designed with a specific zirconia: Zirkolith. As well as the surface treatment by SLM (selective laser melting), this treatment is also patented by the company. The image in Figure 6 shows a fairly clean surface, with a roughness that is characteristic of this type of process.

### 3.3. SEM-EDS Composition

This implant was supposed to be designed with alumina-reinforced zirconia. The manufacturer does not disclose the presence of yttrium. The spectrum of the chemical composition, as in Figure 7, confirms the absence of this element. Table 3 shows the specific composition of each chemical element. This manufacturer has not specified the exact composition of this implant.

The Ziraldent implant from Metoxit^®^ has a patented zircopore surface treatment. This treatment corresponds to a thermohydrolysis of zirconium salt in solution which allows obtaining a microporous surface. The spectrum of the composition of the Ziraldent implant confirms the manufacturer’s data by the presence of aluminum, yttrium, and zirconium up to more than 99.5% (Figure 8). Table 4 shows the comparative composition of chemical elements present in the implant. This manufacturer has not specified the exact proportions of each chemical element.

The images obtained, as in Figure 9, reveal a surface that seems clean and without impurities. On the other hand, the composition reveals the presence of approximately 15% aluminum, 75% zirconium, and 10% yttrium. However, this does not correspond to the proportions given by the manufacturer (Table 5).

## 4. Discussion

This study provides relevant information on the strategies of designers to propose competitive products according to their respective specifications. Thus, the objective is to compare the surface conditions and the intrinsic compositions of three manufacturers of zirconia implants, and the results of this work put into perspective the legitimate questioning of consumers on the durability of certain medical devices. The Ra values measured on the implants under study range from 0.3 µm for NobelPearl^®^ to 0.8 µm for the Z-Systems^®^ Z look 3. These values are well below the values announced by the manufacturers, which are in the order of 1 to 2 µm of Ra. It should be noted that these results are correlated with the SEM images of the surface obtained. However, it is quite legitimate to wonder why there is such a difference between the data of the parameters announced by the manufacturers and the real parameters of the marketed devices. A commercial ethic must be respected given the requirements of the clinicians who prefer to place either implants with a roughened surface or implants with a machined surface according to their own clinical experience. Indeed, it is known that the surface topography of the implant conditions in sine qua non was in the process of osseointegration [17]. More explicitly, recent studies in implanted dogs have shown that during bone healing, verticalization reactions occur, leading to the formation of cortical bone around implants with a machined surface, in contrast to trabeculation reactions around implants with a rough surface [18,19]. Thus, osteogenic differentiation is a function of the topographic surface orientation, which conditions the primary stability and/or longevity of the implant placed [18]. Some clinicians prefer to place machined implants rather than rough implants because of the good cortical bone anchorage, which leads to solid implant stability. Given this, manufacturers must communicate transparently about the technical parameters of the implants marketed. In the same vein, a recent study conducted by our team to compare the surface conditions and composition of several titanium implants revealed the absence of an independent recognized standard for the surface conditions of dental implants and the lack of independent quality control by the manufacturers [20]. In addition to this, the surface conditions also play a role in the colonization of the bacterial biofilm which can lead to corrosion of the material, peri-implantitis, and pre-implant necrosis, which together constitute the main cause of implant failures with a prevalence that can go up to 56% [21]. More relevantly, a large systematic review that analyzed more than 4700 articles or 700 implants statistically demonstrated that the rate of peri-implantitis depends on surface treatments. Thus, machined surfaces with an Sa of less than 0.9 µm have a prevalence of only 0.56%, while rough surfaces with an Sa of more than 2 µm have a prevalence of 12.86% [22]. These clinical observations are all the more alarming because clinicians are turning back the clock and increasingly favoring the placement of machined surface finish implants.

The results of the intrinsic composition of zirconia implants in this study highlight some disturbing observations. SEM-EDS analysis of the chemical composition spectra of the zirconia implants suggests that the Z-Systems^®^ Z look 3 implants are composed of 14.37% aluminum not declared by the manufacturer. This disturbing finding suggests that manufacturers are not subject to an independent control system of marketed dental implants that could sanction irregularities [20]. Similarly, analysis of the chemical composition spectra suggests that Metoxit^®^’s Ziraldent FR2 is composed of 42.43% aluminum, and Z-Systems^®^’s Z look 3 is composed of 43.73% aluminum. These respective manufacturers have announced the presence of aluminum in their implants, but they do not indicate the exact proportions of aluminum. Certain, manufacturers mention the better resistance to aging and increased toughness of zirconia matrix composite ceramics, although in principle, they are meant to be composed of only 20% aluminum which is very far from being the case in this study. It should be noted that these proportions are high compared to the aluminum proportions of 4% for TiAl6V used to improve the mechanical properties of titanium or the 3% or 5% of yttrium oxide necessary to stabilize the zirconia. It is essential to understand that the objective of this type of strategy is often economic, given the costs of raw materials. It is legitimate to think about increasing the mechanical properties of materials by reasonably modulating the content of chemical elements, such as aluminum. However, it is more important to consider the potential health risks in case of systemic release of these aluminum particles into the body of implanted patients. As far as patient health is concerned, TiAl6V is the reference material in dental implantology, but the neutrality of its composition is increasingly being questioned. With clinical hindsight, studies report the dissolution of vanadium and aluminum after implant corrosion in vivo [23]. Indeed, vanadium has a cytotoxic effect and aluminum has significant neurotoxic effects and can induce peri-implant osteolysis and local inflammation, as well as Alzheimer’s disease by exposure of the brain to systemically released aluminum particles [24,25].

Aluminum is a prime suspect because of its involvement in a variety of adverse health situations, although no physiological function of this metal is known. The systemic release of aluminum from corroded titanium implants may cause inhibition of peri-implant bone formation and mineralization by increasing osteoclastic activity while reducing osteoblastic activity [26,27]. In the same logic, a study suggests the involvement of aluminum in the induction of cell apoptosis, erythrose, and tissue necrosis [28,29]. Aluminum can also induce alterations in mitochondrial bioenergetics and can lead to the generation of oxidative stress including lipid peroxidation [30]. Aluminum would be involved in various important physiological dysfunctions ranging from neurodegeneration to endocrine disruptors while passing by major inflammatory dysfunctions that can attack a multitude of various organs including ischemic strokes endangering the life of the exposed individuals [30]. In the same sense, a study established a direct correlation between the appearance of Alzheimer’s disease as well as other encephalopathies and the important content of aluminum in the cerebral tissue [31]. It has also been reported that aluminum in the hippocampus and glial cells of some patients was responsible for the development of late-onset epilepsy [32]. Cognitive and neurobehavioral disorders associated with aluminum exposure-related neuropathologies have been studied [33]. The results of these studies speak for themselves given the seriousness of the health risks associated with aluminum toxicity. However, additional studies on a larger number of implants from different manufacturers are needed to make an accurate overall assessment. Thus, regulations involving compliance studies and independent quality control should be put in place to systematically correct irregularities in designers’ practices.

## 5. Conclusions

The results of this study, which compared the surface finishes and intrinsic compositions of three zirconia implant manufacturers, have brought to light a concerning aspect. The findings revealed a significant use of aluminum as a chemical substitute by manufacturers to optimize production. Furthermore, some manufacturers do not disclose the presence of this element in their implants, while others lack full transparency regarding the proportions of aluminum used, potentially exposing patients to proven health risks.

It is important to acknowledge that these observations are indeed concerning. However, it is essential to recognize that these implants can be improved, and from a technical perspective, the brands have room for enhancement. By addressing these issues and striving for better transparency and safety standards, manufacturers have the opportunity to provide even more reliable and advanced products to patients. Furthermore, several research directions are being pursued to enhance aluminum-doped zirconia dental implants. Researchers have the potential to optimize the composition of aluminum-doped zirconia to reduce aluminum release while preserving the mechanical properties and biocompatibility of the implant. It is also feasible to develop special surface coatings to seal aluminum-doped zirconia, thereby preventing direct contact between aluminum and biological tissues. Thorough studies on aluminum release from these implants under simulated physiological conditions are crucial to better understand aluminum behavior and develop effective containment solutions. Furthermore, long-term clinical studies are essential to assess the stability of aluminum-doped zirconia implants and monitor any adverse effects on patient health. Research into alternative materials for “metal-free” dental implants can also be undertaken to completely eliminate the use of aluminum. Finally, regulatory bodies and manufacturers must establish monitoring and control mechanisms to ensure that implants comply with safety and reliability standards.

## Figures and Tables

**Figure 1 bioengineering-10-01102-f001:**
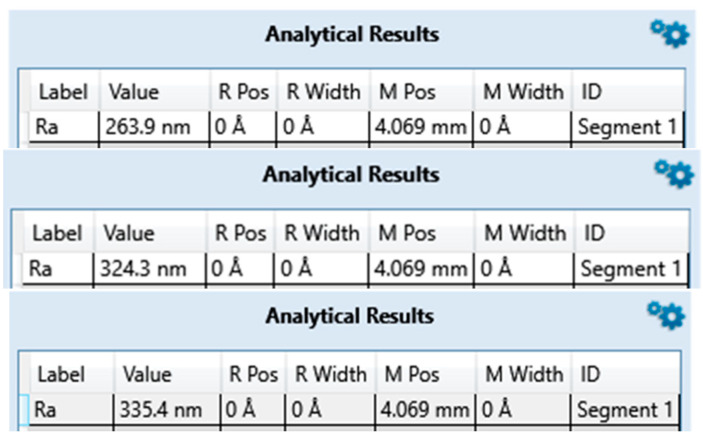
Surface roughness measured for ZrO2-1 exported from Bruker DektakXT stylus^®^ Vision 64.

**Figure 2 bioengineering-10-01102-f002:**
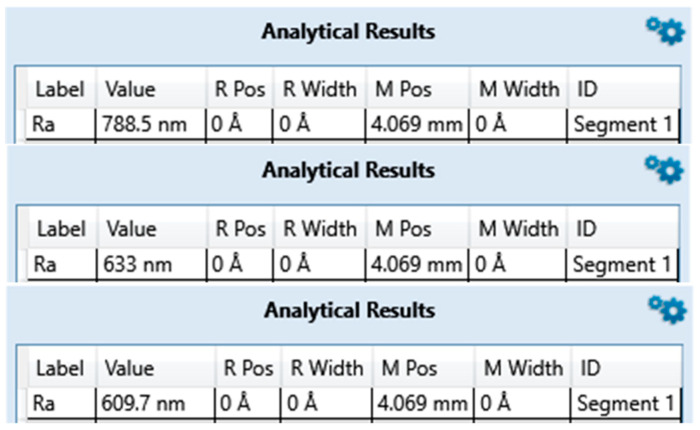
Surface roughness measured for ZrO2-2 exported from Bruker DektakXT stylus^®^ Vision 64.

**Figure 3 bioengineering-10-01102-f003:**
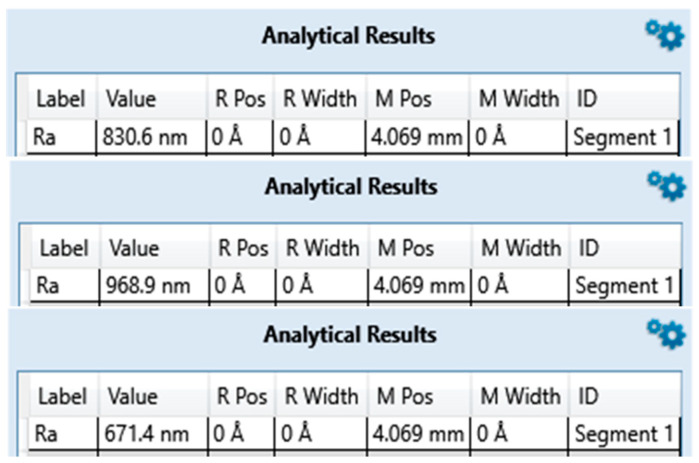
Surface roughness measured for ZrO2-3 exported from Bruker DektakXT stylus^®^ Vision 64.

**Figure 4 bioengineering-10-01102-f004:**
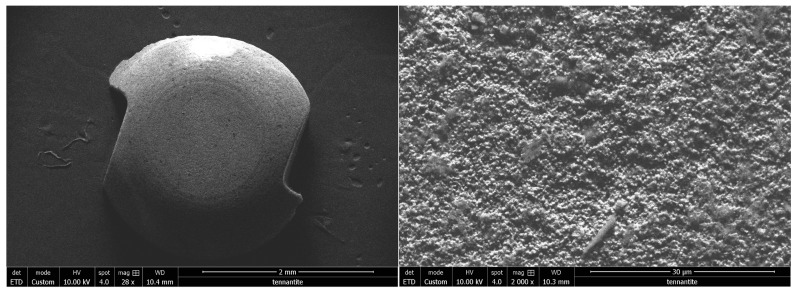
Scanning electron microscope observation of Nobelpearl (ZrO2-1) of the Nobel^®^ brand implant at ×28 magnification on the left and ×2000 on the right.

**Figure 5 bioengineering-10-01102-f005:**
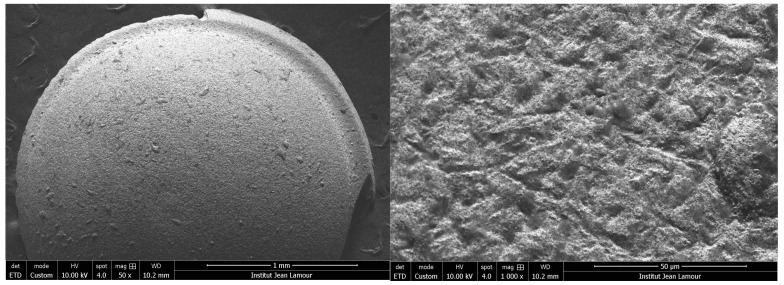
Scanning electron microscope observation of the Ziraldent (ZrO2-2) of the Metoxit^®^ brand implant at ×50 magnification on the left and ×1000 on the right.

**Figure 6 bioengineering-10-01102-f006:**
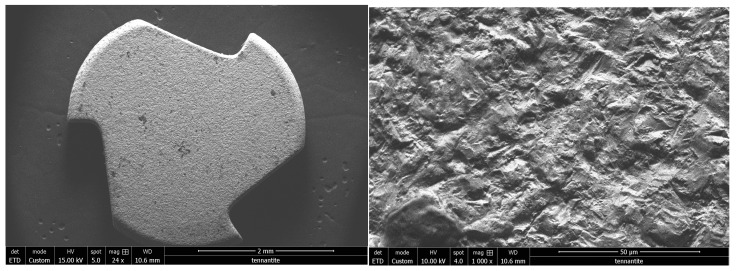
Scanning electron microscope observation of the Z look 3 (ZrO2-3) of the Z-systems^®^ brand implant at ×24 magnification on the left and ×1000 on the right.

**Figure 7 bioengineering-10-01102-f007:**
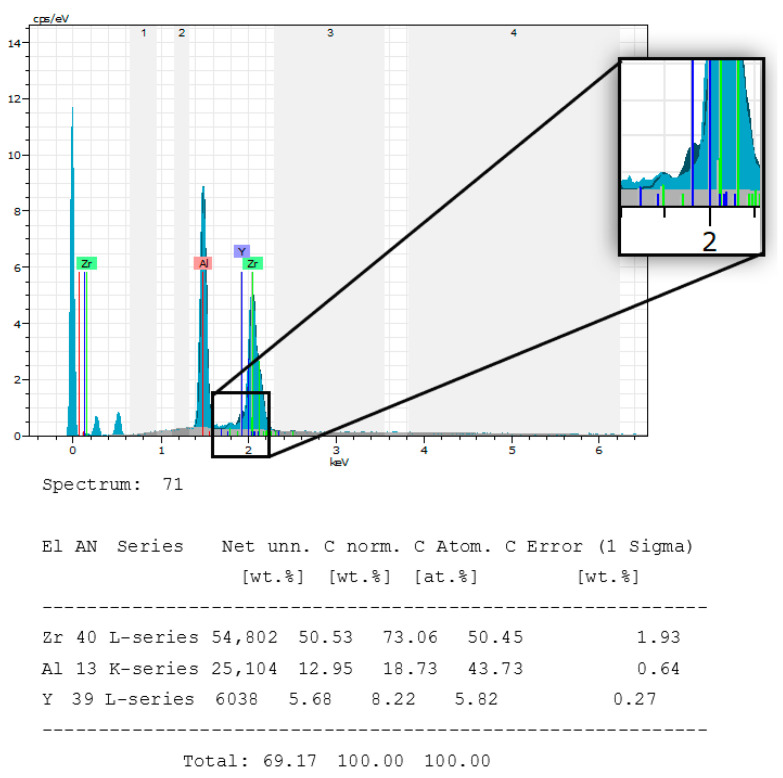
Chemical composition spectrum of the Nobelpearl (ZrO2-1) of the Nobel^®^ brand implant sample. The absence of yttrium is represented by a solid dark blue area inside the frame.

**Figure 8 bioengineering-10-01102-f008:**
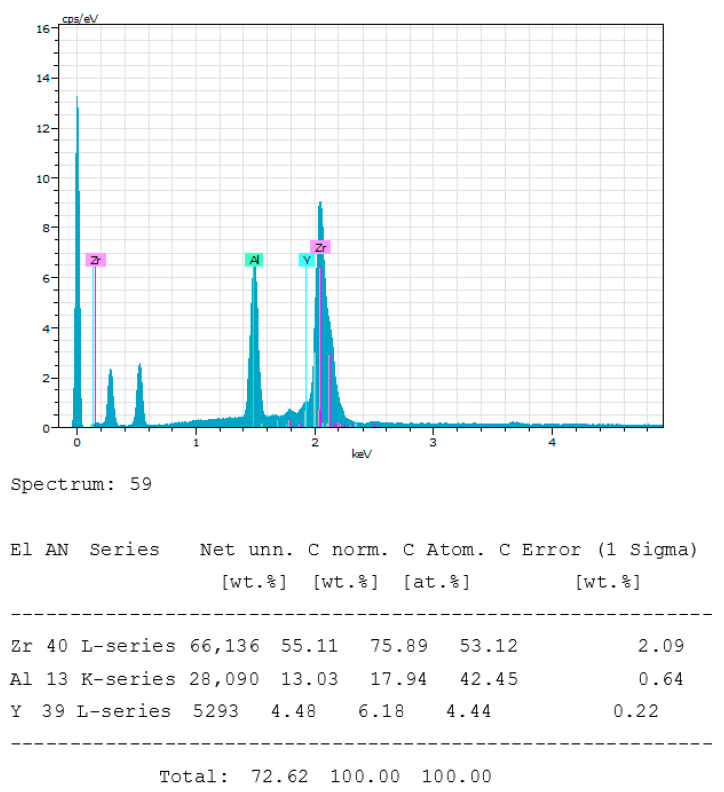
Chemical composition spectrum of the Ziraldent (ZrO2-2) Metoxit^®^ implant sample.

**Figure 9 bioengineering-10-01102-f009:**
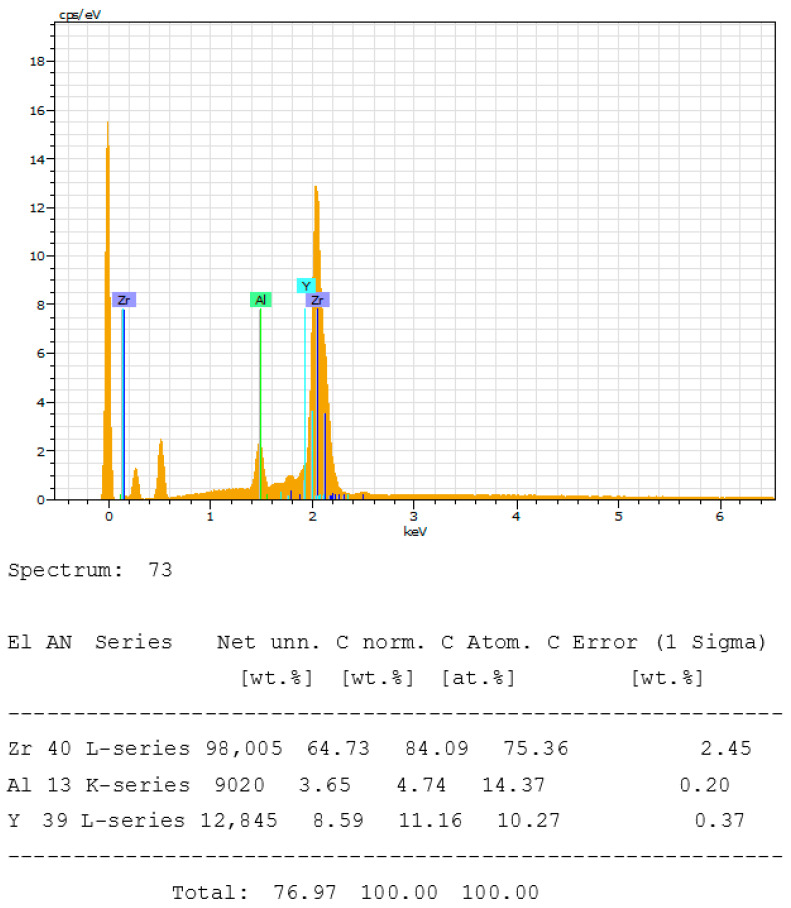
Spectrum of the chemical composition of the Z look 3 (ZrO2-3) of the Z-systems^®^ brand implant sample.

**Table 1 bioengineering-10-01102-t001:** Implant brands and models.

	ZrO2-1	ZrO2-2	ZrO2-3
**Brands**	Nobel^®^	Metoxit^®^	Z-systems^®^
**Model**	NobelPearl	Ziraldent FR2	Z look 3 Evo
**Dimensions**	4.2 × 10	4 × 9	5 × 13

**Table 2 bioengineering-10-01102-t002:** Average roughness results of zirconia implant samples.

	ZrO2-1	ZrO2-2	ZrO2-3
**Brands**	Nobel^®^	Metoxit^®^	Z-systems^®^
**Model**	NobelPearl	Ziraldent FR2	Z look 3 Evo
**Ra (µm) Announced**	Not specified	2	3.62
**Ra (µm) Measured**	0.307	0.677	0.823

**Table 3 bioengineering-10-01102-t003:** Comparison between specified and measured composition.

Nobelpearl (ZrO2-1)	ZrO_2_	Al_2_O_3_	Y_2_O_3_
Reported composition	Not specified		
Measured composition	50.45%	43.73%	5.82%

**Table 4 bioengineering-10-01102-t004:** Comparison between specified and measured composition.

Ziraldent FR2 (ZrO2-2)	ZrO_2_	Al_2_O_3_	Y_2_O_3_
Reported composition	>99.5%		
Measured composition	53.12%	42.45%	4.44%

**Table 5 bioengineering-10-01102-t005:** Comparison between specified and measured composition.

Z Look 3 (ZrO2-3)	ZrO_2_	Al_2_O_3_	Y_2_O_3_
Reported composition	95%	0.25%	5%
Measured composition	75.36%	14.37%	10.27%

## Data Availability

Data sharing not applicable.

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
