# Peer review of "Zirconia Dental Implants: A Closer Look at Surface Condition and Intrinsic Composition by SEM-EDX"

_bioengineering, 2023, doi:10.3390/bioengineering10091102_

Round 1

Reviewer 1 Report

In this manuscript, “Zirconia Dental Implants: A Closer Look at Surface Condition 2 and Intrinsic Composition by SEM-EDX”, the authors compared the surface finishes and intrinsic composition of three zirconia implants from three major brands. Besides, the authors demonstrated that many dental manufacturers utilized aluminum as a chemical substitute without mentioning the presence of doctors or patients. The idea of the paper is essential and interesting. But many problems must be clarified before publishing in the Journal of Bioengineering.

Q1. On page 2, line 91, the authors explain that they used a micro-saw (Struers secotom-10) to get the cross-sections of each implant. The authors need to give a detailed explanation to see why the cutting wheel materials of secotom-10 will not impact SEM-EDS composition analysis.

Q2. On page 3, Table 2, the authors measured the average roughness of zirconia implant samples. And the results showed to be contrary to expectations. Please give the raw data (e.g., from Bruker’s Vision64 software) and add more locations to show the correct data collection process.

Q3. In Table 3-5, the authors gave the composition list of ZrO2, Al2O3, and Y2O3. Please merge them into one table and give the calculated process of how the authors changed the EDS-measured composition to the data in the tables.

Q4. The authors need to give more data to certify that Al elements existed on three implants like XPS.

None

Reviewer 2 Report

In this manuscript, authors reported on "Zirconia Dental Implants: A Closer Look at Surface Condition and Intrinsic Composition by SEM-EDX". The hypothesis of the study meets the quality of the journal, which I recommended for publication. However, authors need to address the below-mentioned points in the revised manuscript.

1. In the abstract, the authors indicated "In addition, some manufacturers do not mention the presence of this element in their implants". What are the reasons manufacturers don’t mention the presence of elements? If possible, please mention that manufacturer's (brand) name in the manuscript.

2. If possible, add information about "what is the guidance for manufacturing Zirconia Dental Implants for commercial use.

3. Include the significant drawbacks of choosing zirconia dental implants.

4. In conclusion, the authors indicate "these implants can be improved" If possible; include some more information on "what are the new innovations and research that will help to improve these types of implants in the health care system". 

moderate
